# Detecting Risk Of Postural hypotension (DROP): derivation and validation of a prediction score for primary care

Christopher Elles Clark,[1] Daniel Thomas,[1] Fiona C Warren,[1] David J Llewellyn,[2] Luigi Ferrucci,[3] John L Campbell[1]

Interim reports on this work have been presented at annual scientific meetings of the European Society for Hypertension, Paris 2016 (Clark C, Thomas D, Warren F et al. Predicting postural hypotension, falls, and cognitive impairment: the InCHIANTI study. J Hypertens 2016; 34, e-Supplement 2: e32, September 2016) and the British and Irish Hypertension Society, Dublin, 2016 (Clark C, Thomas D, Mejzner N, et al. Can we predict who should be tested for postural hypotension? Derivation and validation of a prediction tool. Journal of Human Hypertension 2016;30 doi: doi:10.1038/jhh.2016.60)

For numbered affiliations see end of article.

**Correspondence to**
Dr Christopher Elles Clark;
c.e.clark@exeter.ac.uk

## ABSTRACT

**Objectives** Falls are a common problem in older people. Postural hypotension contributes to falls but is often asymptomatic. In the absence of symptoms, postural hypotension is only infrequently checked for in clinical practice. We undertook this study to derive, validate and explore the prospective associations of a prediction tool to identify people likely to have unrecognised postural hypotension.

**Design and setting** Cross-sectional and prospective multivariable cohort analysis.

**Participants** 1317 participants of the Invecchiare in Chianti study, a population-based cohort representative of the older Italian population.

**Primary outcome measures** Predictive value of score to suggest presence of postural hypotension.

**Methods** Subjects were randomised 1:1 to derivation or validation cohorts. Within the derivation cohort, univariable associations for candidate predictors of postural hypotension were tested. Variables with p<0.1 entered multivariable linear regression models. Factors retaining multivariable significance were incorporated into unweighted and weighted Detecting Risk Of Postural hypotension (DROP) scores. These scores were tested in the validation cohort against prediction of postural hypotension, cognitive decline and mortality over 9 years of follow-up.

**Results** Postural hypotension was present in 203 (15.4%) of participants. Factors predicting postural hypotension were: digoxin use, Parkinson's disease, hypertension, stroke or cardiovascular disease and an interarm systolic blood pressure difference. Area under the curve was consistent at 0.65 for all models, with significant ORs of 1.8 to 2.4 per unit increase in score for predicting postural hypotension. For a DROP score ≥1, five cases need to be tested to identify one with postural hypotension. Increasing DROP scores predicted mortality (OR 1.8 to 2.8 per unit rise) and increasing rates of decline of Mini Mental State Examination score (analysis of variance p<0.001) over 9 years of follow-up.

**Conclusions** The DROP score provides a simple method to identify people likely to have postural hypotension and increased risks to health who require further evaluation.

## Strengths and limitations of this study

► This study used data from a well-established cohort representative of an older population in Italy to derive and validate a score (Detecting Risk Of Postural hypotension (DROP) score) to predict the presence of postural hypotension.

► Comprehensive recording of baseline variables at recruitment by the Invecchiare in Chianti (InCHIANTI) investigators allowed a large number of previously reported risk markers for postural hypotension to be tested in the analyses.

► The study was undertaken according to the Transparent Reporting of a multivariable prediction model for Individual Prognosis Or Diagnosis (TRIPOD) statement and randomised splitting of the cohort allowed internal validation of the findings to be undertaken.

► We chose the consensus definition of postural hypotension as our outcome measure since we sought to predict this rather than study postural symptoms. Specific postural symptoms were not recorded during recruitment to the InCHIANTI study, and their presence should in any event trigger testing for postural hypotension.

► The population studied did not include residential or nursing home residents; refinement of the scoring system within larger cohorts more representative of primary care populations is required to confirm the potential of the DROP score in practice.

## INTRODUCTION

Falls are a major cause of morbidity and mortality in older people; 35% of people older than 65% and 50% of people older than 80 fall at least once a year.[1 2] Falls are the leading cause of disability and the leading cause of death from injury among people over 75 in the UK, and cost the National Health Service around £2.3 billion per year.[3] Postural or orthostatic hypotension is a major risk factor for falls[4 5] and is independently associated with increased mortality rates.[6–8] Postural hypotension has also been associated with dementia and cognitive impairment and may have more subtle adverse effects on well-being and cognition.[9]

Postural hypotension is commonly defined as a fall of either ≥20 mm Hg in systolic blood

pressure or ≥10 mm Hg in diastolic blood pressure, from sitting or lying, within 3 min of standing up.[10] Reported prevalences of postural hypotension vary widely and are sensitive to both care setting, occurring in over half of patients admitted to care of the elderly[11–13] and to the presence of comorbidity. General adult population prevalence appears to be around 7%,[14 15] rising to 11% to 15% in persons 65 years old and older[16–18] and 19% in those aged over 80 or older.[15] Prevalence is reported to be higher in the presence of hypertension,[19–23] stroke,[24 25] myocardial infarction[25 26] and diabetes.[22 27]

Guidelines vary in recommendations for the detection of postural hypotension. The National Institute for Health and Care Excellence recommends testing in the presence of symptoms whilst the European Society for Hypertension also recommends testing in the elderly and in the presence of diabetes.[1 28] Unfortunately, most individuals with postural hypotension are asymptomatic,[7] and we have found that, in practice, postural hypotension is seldom looked for in patients who do not report postural symptoms.[29] Anecdotally, testing is not undertaken due to time constraints; screening for postural hypotension is not supported in the literature, being regarded as lacking an evidence base, and primary care workloads are rising.[30 31] Risks of hospitalisation, nursing home admission or mortality can already be predicted by the electronic frailty index (eFI), a score derived from existing information in primary care computer records and incorporated into many general practice computing systems. However, the association of eFI with, and its ability to predict, postural hypotension (which itself is poorly tested for and recorded in primary care) is unclear,[32] and comparable frailty indices have not been found to be predictive of postural hypotension.[33] To address this gap in care, we hypothesised that a simple prediction score, based on easily recognised risk markers, might help clinicians identify those most likely to have postural hypotension thereby allowing a targeted implementation of sitting and standing blood pressure measurement in the absence of symptoms. We therefore undertook the current analysis in a well-documented cohort known to be representative of an older population living in the community. Aims were to explore the feasibility of deriving and internally validating a prediction score to assess its value and its prospective associations.

## METHODS

The study was conducted and reported in accordance with the TRIPOD statement.[34] We studied participants from the Invecchiare in Chianti (InCHIANTI) study; a cohort study designed to explore declining mobility in later life. The Italian National Research Council on Aging Ethical Committee approved the InCHIANTI study protocol, and the current analysis proposals were approved by the investigating committee of the InCHIANTI study.

The InCHIANTI study methods have been described in detail elsewhere.[35] In brief, 1270 participants aged 65 years or more were randomly selected from the population registries of two villages: Greve in Chianti and Antella in Bagno a Ripoli. Additional people were randomly selected from these sites to complete recruitment of at least 30 men and 30 women for each age decile from age 20 to 29 upwards. Extensive baseline interviews and examinations were conducted at recruitment, between September 1998 and March 2000, and follow-up data were obtained after 3, 6 and 9 years. Blood pressure was initially measured supine, sequentially in both arms, to identify the higher reading arm, then a further two measurements were made on the higher reading arm. Subjects then stood and blood pressure was measured once after 1 min and once more after 3 min standing. All measurements were obtained by research assistants using a standard mercury sphygmomanometer. Written informed consent was obtained from all participants at recruitment to the InCHIANTI study.

Baseline blood pressure was calculated as the mean of the second and third supine blood pressure readings.[36] Postural changes in blood pressure from lying to standing were calculated by subtraction of this mean from the standing blood pressure. Postural hypotension was considered to exist where there was as a reduction in blood pressure on standing of ≥20 mm Hg systolic or ≥10 mm Hg diastolic after 1 or after 3 min.[10] Hypertension was defined as use of antihypertensive drugs and/or a documented history of hypertension at recruitment.

For this analysis, participants were randomly allocated in a 1:1 ratio using a split-sample method,[37] stratified for gender and study site, to either a derivation or a validation group by a statistician (FW) blinded to postural hypotension status and medical history. A literature review was undertaken to identify potential risk markers for consideration in the analyses (see online supplementary appendix 1). These were mapped to variables available in the InCHIANTI dataset (table 1), which were then tested in the derivation cohort for univariable associations with postural hypotension, using t-tests or $\chi^2$ tests as appropriate to the data. Variables signalling potential univariable associations (defined as p<0.1) were included in multivariable model analyses using an automated backward stepwise regression method.[38] We also included age (explored both continuously and as a dichotomous variable with cut-offs of 60, 65 and 70 years) and gender in all multivariable models. Prospective associations of postural hypotension with survival up to 9 years of follow-up were tested using Kaplan-Meier plots and Cox proportional hazard ratios. Cognitive decline was defined as a reduction in Mini Mental State Examination score (MMSE score) of five points or more from baseline and rate of cognitive decline was defined as change in MMSE scores averaged per year of follow-up.

Risk markers that retained significance in the multivariable models were used to derive both weighted and unweighted scores (Detecting Risk Of Postural hypotension (DROP) scores); weighted scores were derived by the addition of the multivariable log (n) OR for each marker

**Table 1** Risk markers included in univariable analysis

| Group | Risk marker included in analysis |
| --- | --- |
| Demographics | Age, gender |
| Medical history | Hypertension<br>Heart failure<br>Myocardial infarction<br>Angina<br>Stroke<br>Diabetes<br>Parkinson's disease<br>Cancer<br>Dementia |
| Examination | MMSE |
| Medications | Antihypertensives<br>Antiarrhythmics<br>Antidepressants<br>Antipsychotics<br>Anxiolytics<br>Anticholinesterase inhibitors |
| Frailty | Hospital admission, fall or weight loss in last 12 months<br>WHO physical disability level<br>ADL disability score |

ADL, activities of daily living; MMSE, Mini Mental State Examination.

present, whereas the unweighted model allocated one point for each risk marker present. Scores were tested in the validation cohort for ability to predict postural hypotension using receiver operating characteristic analysis, to predict future mortality using Cox proportional hazard ratios, and cognitive decline over 9 years using analysis of variance. All analyses were undertaken using IBM SPSS Statistics V.24.0.0.2.

## RESULTS

Data for standing blood pressure existed for 1317 of the 1453 participants (91%), and they formed the cohort for this study. The derivation cohort (n=649) and validation cohort (n=668) were well matched for all important characteristics and putative risk markers (table 2); overall postural hypotension was present for 203 (15.4%) of participants at recruitment. Mean age of participants was 68.3 (SD 15.5).

For the derivation cohort, postural hypotension was associated, over 9 years of follow-up, with increased all-cause mortality (HR 1.9; 95% CI 1.4 to 2.7), cardiovascular mortality (HR 2.1; 95% CI 1.2 to 3.4) and non-cardiovascular mortality (HR 2.0; 95% CI 1.3 to 3.0). Results of univariable testing are summarised in table 3. Using a cut-off value of p<0.1 the following candidate predictors were entered into multivariable models: age (continuous or dichotomous for age 60 or 70 cut-offs), MMSE score, angiotensin 2 antagonist, diuretic and digoxin use, presence of hypertension, any cardiovascular disease (composite of history of myocardial infarction, angina

pectoris or congestive heart failure), stroke, Parkinson's disease, hospital admission within the last year, WHO disability level, any disability in activities of daily living and systolic interarm difference (continuous or using ≥10 mm Hg cut-off).

Terms for systolic and diastolic blood pressure were entered into the multivariable model in a sensitivity analysis. Apart from finding that systolic blood pressure replaced the term for presence of hypertension, model outputs were unchanged. Therefore, we adopted the latter for consistency with our aim to derive a pragmatic score.

Backward stepwise regression analysis produced consistent findings with any permutation of discrete and continuous variables for age (which was not retained in any model) or for interarm difference (model 1 and model 2; table 4). Consequently, a dichotomous cut-off for interarm difference of ≥10 mm Hg was selected for simplicity and retained with five other factors (use of digoxin, Parkinson's disease, previous stroke, previous cardiac disease and diagnosis of hypertension) to derive weighted (using log OR) and unweighted (score 1 for each factor present; possible range 0 to 6) DROP scores. The scores were tested in the validation cohort. Since interarm difference is not routinely measured, a third model excluding interarm difference (model 3, table 4) was also used to derive DROP scores without this term (possible range 0 to 5).

All versions of the DROP score were found to predict postural hypotension in the validation cohort with similar areas under the curve of 0.65 but a trend to higher odds of postural hypotension with the exclusion of interarm difference from the model (figure 1, table 5). Sensitivities and specificities of the unweighted DROP score without the interarm difference term were 76%, 16%, 5% and 53%, 91%, 99%, respectively, for cut-offs of ≥1, ≥2 and ≥3, although only 15 participants attained a DROP score of 3 and only one a score of 4. This equated to a number needed to test to detect one case of postural hypotension of 5, 5 and 2 for DROP scores of 1, 2 and 3, respectively. For the weighted DROP score without interarm difference a cut-off value of 0.6 or more had a sensitivity of 74% and specificity of 55% for detection of postural hypotension. A similar pattern was seen for the DROP models including interarm difference; for an unweighted DROP score of 1 or more sensitivity and specificity for postural hypotension were 81% and 46%, respectively, predicting detection of one case of postural hypotension for every five tested. For the weighted score, a cut-off value of 0.26 had a sensitivity of 81% and a specificity of 46% for detection of postural hypotension.

DROP scores were predictive of mortality over 9 years of follow-up, with increasing ORs according to DROP score with adjustment for age (figure 2). Data on MMSE were available for 529/668 (79%) of the validation cohort; classification by unweighted DROP scores was also predictive of decline in MMSE after 9 years (figure 3). DROP scores were not predictive of future falls; however, increasing

**Table 2** Baseline characteristics of derivation and validation cohorts

| N | Derivation cohort 649 Mean (SD) or N/% | Validation cohort 668 Mean (SD) or N/% | P value t/$\chi^2$ |
|---|---|---|---|
| Age | 68.5 (15.7) | 68.2 (15.3) | 0.77 |
| BMI | 27.2 (4.3) | 27.1 (4.0) | 0.59 |
| Supine SBP (higher arm)* | 145.9 (21.3) | 146.3 (21.6) | 0.76 |
| Supine DBP (higher arm)* | 82.9 (8.8) | 83.1 (9.5) | 0.59 |
| Standing SBP 1 min | 140.4 (21.0) | 141.2 (21.3) | 0.51 |
| Standing DBP 1 min | 83.0 (8.9) | 83.6 (9.4) | 0.25 |
| Standing SBP 3 min | 141.4 (20.9) | 141.9 (20.9) | 0.66 |
| Standing DBP 3 min | 82.7 (9.0) | 83.0 (9.4) | 0.60 |
| Female | 368 (56.7) | 358 (53.6) | 0.27 |
| Site (Greve vs Bagno a Ripoli) | 320 vs 329 | 327 vs 341 | 0.91 |
| Deceased at 9 years | 199 (30.7) | 203 (30.4) | 0.95 |
| Systolic drop ≥20 mm Hg 1 min | 56 (8.6) | 45 (6.7) | 0.21 |
| Diastolic drop ≥10 mm Hg 1 min | 41 (6.3) | 40 (6.0) | 0.82 |
| Systolic drop ≥20 mm Hg 3 min | 47 (7.2) | 42 (6.3) | 0.51 |
| Diastolic drop ≥10 mm Hg 3 min | 46 (7.1) | 48 (7.2) | 1.00 |
| Postural hypotension present† | 107 (16.5) | 96 (14.4) | 0.32 |
| Systolic interarm difference ≥10 mm Hg | 121 (18.8) | 121 (18.1) | 0.83 |
| Previous stroke | 44 (6.8) | 45 (6.7) | 1.00 |
| Pre-existing diabetes | 80 (12.3) | 76 (11.4) | 0.61 |
| Pre-existing hypertension | 279 (43.0) | 292 (43.7) | 0.82 |
| Pre-existing CV disease | 63 (9.7) | 50 (7.5) | 0.17 |
| Pre-existing dementia | 38 (5.9) | 27 (4.0) | 0.16 |
| Pre-existing Parkinson's disease | 9 (1.4) | 6 (0.9) | 0.45 |
| Fall in preceding 12 months | 143 (22.0) | 130 (19.5) | 0.28 |

*Mean of second and third readings.
†Defined as a drop of ≥20 mm Hg systolic or ≥10 mm Hg diastolic within 3 min of standing.
BMI, body mass index; CV, cardiovascular; DBP, dystolic blood pressure; SBP, systolic blood pressure.

DROP scores were associated with rising prevalence of falls in the year prior to recruitment ($\chi^2$ for trend p<0.001).

## DISCUSSION
### Main findings
This analysis has confirmed that it is feasible, in a community living cohort of predominantly older people, to derive a score based on easily recognised risk markers that can help to identify older persons that are likely to have postural hypotension and require further clinical evaluation. The score, consisting of six risk markers (use of digoxin, presence of Parkinson's disease, hypertension, cardiovascular disease, stroke and a difference in systolic blood pressure between arms≥10 mm Hg), performs similarly with or without weighting, therefore a simple additive score is preferred. Performance is also similar when the interarm term is omitted, further simplifying its application.

In this population, postural hypotension is associated with a doubling of risk of death over 9 years of follow-up. The DROP score also predicts increasing future mortality from any cause and is associated with greater decline in MMSE scores.

### Strengths and weaknesses
The cohort was chosen as representative of a free-living elderly population and the 15.4% prevalence of postural hypotension is consistent with figures ranging from 11% to 15% in other general elderly (over 65) populations.[16–18] Comprehensive recording of baseline variables allowed a large number of previously reported risk factors for postural hypotension to be tested. Since this was undertaken as a feasibility study, no formal sample size calculation was undertaken; however, there were sufficient events to support the multivariable analyses performed.[38] Although the relatively low numbers attaining DROP scores higher than 2 did lead to imprecision around the predictive values of those higher levels

**Table 3**  Univariable associations of risk markers with postural hypotension in derivation cohort

| Variable (n (%) unless otherwise stated) | PH absent (n=542) | PH present (n=107) | P value |
|---|---|---|---|
| Age (mean, SD) | 67.7 (15.8) | 72.2 (14.6) | 0.005 |
| Age over 60 | 438 (81) | 96 (90) | 0.027 |
| Age over 65 | 421 (78) | 90 (84) | 0.160 |
| Age over 70 | 302 (56) | 73 (68) | 0.018 |
| MMSE score (mean, SD) | 25.3 (4.9) | 24.1 (5.1) | 0.031 |
| Female gender | 301 (55.5) | 67 (62.6) | 0.200 |
| ACE inhibitors | 103 (19) | 23 (22) | 0.552 |
| Angiotensin-2 antagonists | 6 (1) | 4 (4) | 0.066 |
| Calcium channel blockers | 62 (11) | 15 (14) | 0.451 |
| Diuretics | 48 (9) | 17 (16) | 0.027 |
| Beta-blockers | 20 (4) | 4 (4) | 0.981 |
| Alpha-blockers | 11 (2) | 1 (1) | 0.442 |
| Aldosterone antagonists | 2 (0.4) | 0 (0) | 0.529 |
| Digoxin | 27 (5) | 14 (13) | 0.004 |
| Antiarrhythmics, class I and III | 10 (2) | 4 (4) | 0.264 |
| Psycholeptics: typical antipsychotics | 8 (1) | 4 (4) | 0.119 |
| Psycholeptics: atypical antipsychotics | 6 (1) | 1 (1) | 1.000 |
| Psycholeptics: anxiolytics | 103 (19) | 18 (17) | 0.684 |
| Psychoanaleptics: antidepressants | 22 (4) | 5 (5) | 0.791 |
| Drugs for dementia | 5 (1) | 0 (0) | 1.000 |
| Hypertension | 217 (40) | 62 (58) | 0.001 |
| Congestive heart failure | 22 (4) | 10 (9) | 0.028 |
| Myocardial infarction | 23 (4) | 6 (6) | 0.607 |
| Angina | 21 (4) | 7 (6) | 0.421 |
| Any CV disease | 45 (8) | 18 (17) | 0.011 |
| Stroke | 28 (5) | 16 (15) | 0.001 |
| Diabetes | 64 (12) | 16 (15) | 0.420 |
| Parkinson's disease | 4 (1) | 5 (5) | 0.008 |
| Any cancer | 30 (6) | 8 (8) | 0.497 |
| Dementia | 29 (5) | 9 (8) | 0.257 |
| MMSE score 22 to 26 | 150 (28) | 27 (25) | 0.637 |
| Hospital admission in past year | 54 (10) | 18 (17) | 0.044 |
| Weight loss >4.5Kg in past year | 22 (4) | 7 (6) | 0.301 |
| Any fall in past year | 115 (21) | 28 (26) | 0.254 |
| Any ADL disability | 100 (19) | 28 (26) | 0.083 |
| WHO disability level >1 | 66 (12) | 24 (23) | 0.045 |
| Systolic blood pressure (mean, SD) mm Hg | 144.3 (20.1) | 153.7 (25.3) | <0.001 |
| Diastolic blood pressure (mean, SD) mm Hg | 82.2 (8.8) | 86.2 (8.1) | <0.001 |
| Systolic interarm difference (mean, SD) mm Hg | 2.0 (4.1) | 4.7 (5.9) | <0.001 |
| Systolic interarm BP difference ≥10 mm Hg | 81 (15) | 40 (37) | <0.001 |
| Systolic interarm BP difference ≥15 mm Hg | 10 (2) | 6 (6) | 0.007 |

P values derived from t-tests for continuous data or Pearson $\chi^2$ for categorical data; Fisher's exact test reported where expected cell count <5.

ADL, activities of daily living; BMI, body mass index; BP, blood pressure; CV. cardiovascular; MMSE, Mini Mental State Examination; PH, postural hypotension.

| Table 4 Multivariable prediction models for postural hypotension | | |
| --- | --- | --- |
| Variable | OR | 95% CI |
| Model 1 | | |
| Parkinson's disease | 4.7 | 1.2 to 19.2 |
| Previous stroke | 2.2 | 1.1 to 4.5 |
| Taking digoxin | 2.2 | 1.0 to 4.7 |
| Previous cardiac disease | 1.9 | 1.0 to 3.6 |
| Hypertension | 1.7 | 1.1 to 2.6 |
| Systolic interarm difference (continuous per mm Hg) | 1.1 | 1.1 to 1.2 |
| Model 2 | | |
| Parkinson's disease | 5.0 | 1.2 to 19.9 |
| Previous stroke | 2.2 | 1.1 to 4.4 |
| Taking digoxin | 2.4 | 1.1 to 5.1 |
| Previous cardiac disease | 1.9 | 1.0 to 2.6 |
| Hypertension | 1.7 | 1.1 to 5.1 |
| Systolic interarm difference ≥10 mm Hg | 3.3 | 2.0 to 5.3 |
| Model 3 | | |
| Parkinson's disease | 5.3 | 1.4 to 20.4 |
| Previous stroke | 2.4 | 1.2 to 4.8 |
| Taking digoxin | 2.0 | 0.9 to 4.3 |
| Previous cardiac disease | 1.8 | 0.9 to 3.4 |
| Hypertension | 1.9 | 1.3 to 3.0 |

of scores. Reanalysis and external validation in a larger sized cohort could overcome this limitation. Blood pressures were measured supine and standing for this study, whereas in practice sitting and standing measurements are commonly recommended.[36] These are less sensitive but more practical in primary care[39]; however, a score derived in supine to standing cases of postural hypotension cannot be assumed to perform similarly in the sitting to standing setting. Therefore, we regard this analysis as a feasibility study that supports the concept of a simple pragmatic prediction score to aid daily practice in need of refinement through larger scale analyses and exploration in cohorts with sit to stand measurements. Although the DROP score was associated with fall prevalence, we did not have data on specific posture-induced symptoms, so we were unable to examine the relationship of the DROP score with postural symptoms. The presence of symptoms, however, should trigger testing for postural hypotension in any event.[1 29]

### Relevance to literature

Postural hypotension has previously been reported as a significant independent predictor of 4-year all-cause mortality in the Honolulu Heart programme.[6] It also predicted mortality in the Malmo Heart study[8] but not in the Helsinki ageing study.[40] Frailty was associated with a higher prevalence of postural hypotension in The Irish Longitudinal Study on Ageing (TILDA) study, and adjustment for frailty may influence associations with mortality.[41 42] However no measures of frailty remained

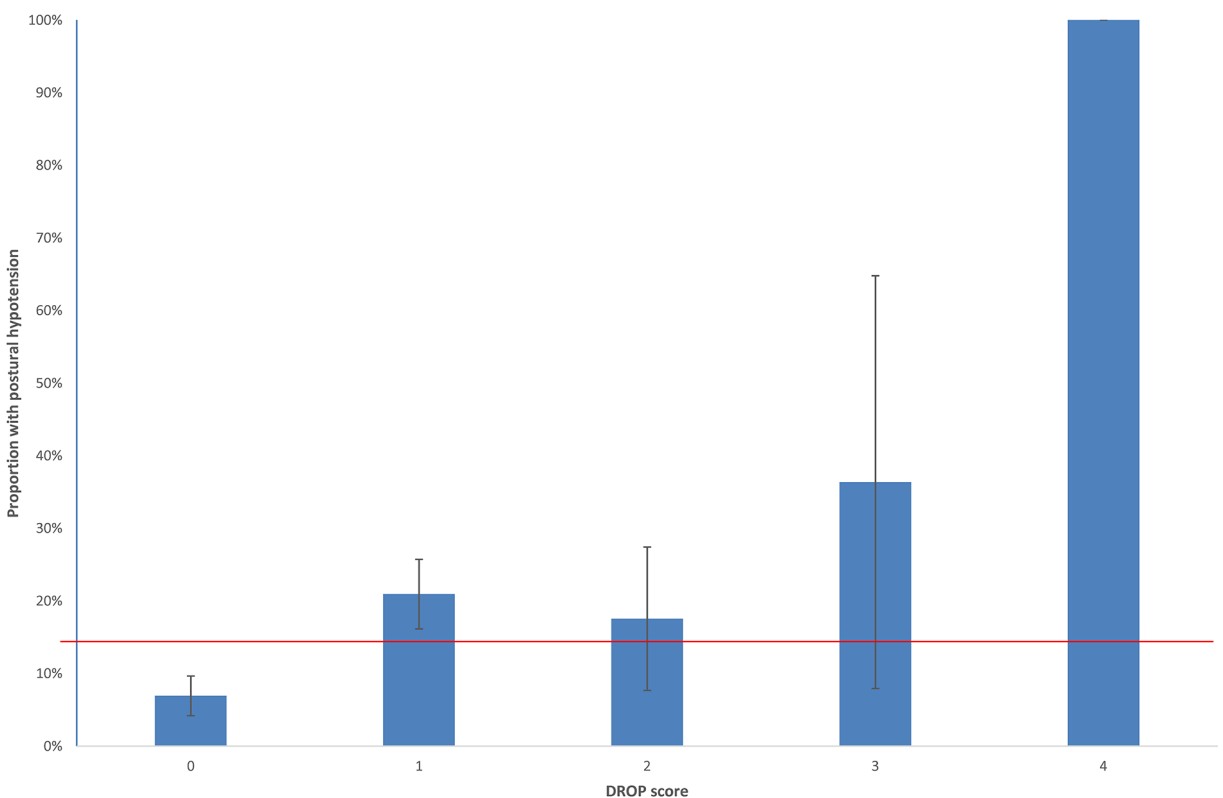

**Figure 1** Prevalence of postural hypotension versus unweighted DROP Score without interarm difference term (population prevalence indicated by horizontal line). DROP, Detecting Risk Of Postural hypotension.

**Table 5** DROP score associations with postural hypotension, mortality and cognitive decline

| | Including interarm difference | | Excluding interarm difference | |
|---|---|---|---|---|
| | **Weighted** | **Unweighted** | **Weighted** | **Unweighted** |
| Prediction of PH per unit increase of DROP score OR (95% CI) | 1.9 (1.4 to 2.5) | 1.8 (1.4 to 2.3) | 2.4 (1.6 to 3.4) | 2.0 (1.5 to 2.6) |
| Area under ROC curve (95% CI) | 0.65 (0.59 to 0.70) | 0.65 (0.60 to 0.71) | 0.65 (0.59 to 0.71) | 0.65 (0.59 to 0.70) |
| Mortality risk per unit score OR (95% CI) | 1.9 (1.6 to 2.2) | 1.8 (1.5 to 2.1) | 2.8 (2.2 to 3.4) | 2.1 (1.8 to 2.5) |
| Change in MMSE score over study (ANOVA) | NA | p=0.004 | NA | p<0.001 |
| Annual change in MMSE score (ANOVA) | NA | p<0.001 | NA | p<0.001 |

ANOVA, analysis of variance; DROP, Detecting Risk Of Postural hypotension; MMSE, Mini Mental State Examination; NA, not applicable; PH, postural hypotension; ROC, receiver operating characteristic.

predictive of postural hypotension on inclusion in the current multivariable analyses, and a frailty index predicted postural *symptoms* but not postural hypotension within TILDA.[33]

Prevalence of postural hypotension rises with age.[15] Although those with postural hypotension in this study were on average 5 years older, age was not a significant independent predictor of postural hypotension in our models. This may have been in part due to the skewed nature of the age profile in InCHIANTI, although sensitivity analyses excluding those under 65 made no difference (not reported). Prevalence of postural hypotension is elevated in association with a history of stroke or transient ischaemic attack,[43–45] cardiovascular disease,[24–26] diabetes[22 27] or hypertension, which itself affects over

60% of the over 65 age group.[46] Thus, the significant factors in our models were all age-related conditions which seems the likely explanation for loss of age itself as an independent predictor due to collinearity. Parkinson's disease was the strongest predictor of postural hypotension in our analyses although, affecting only 1.1% of participants, it was also the least common factor. Postural hypotension has previously been reported to have prevalence approaching 50% in some groups of Parkinson's sufferers,[47 48] although only a third of those with postural hypotension report symptoms.[49]

The association of postural hypotension with presence of an interarm difference is, to our knowledge, a novel finding. We have previously associated interarm difference with white coat effects, which can confound detection of

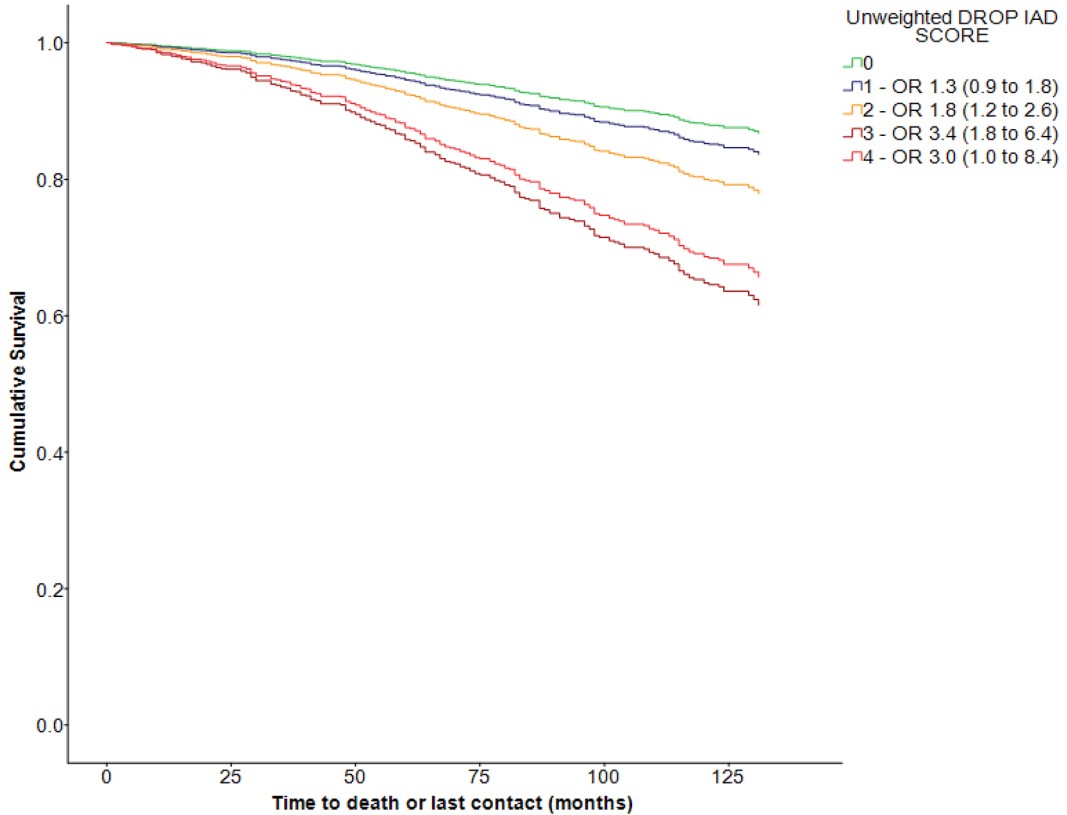

**Figure 2** Kaplan-Meier survival plot for DROP scores over 9 years follow-up. DROP, Detecting Risk Of Postural hypotension; IAD, Inter-arm difference.

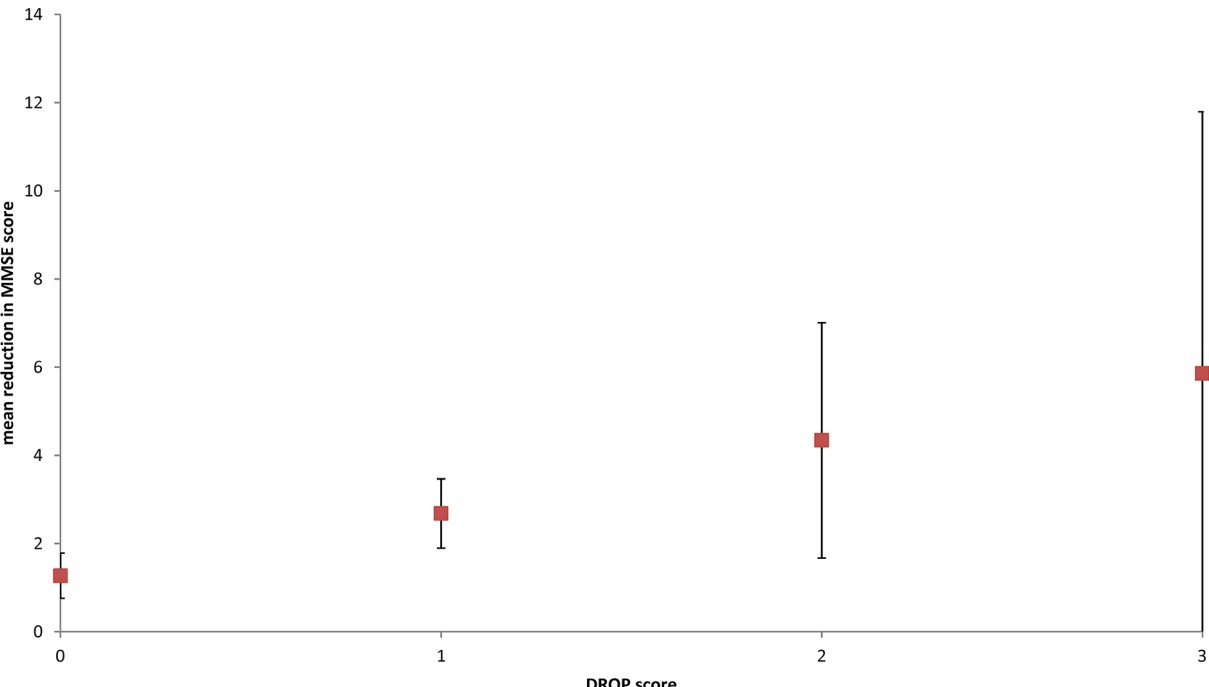

**Figure 3** Mean change in MMSE score over 9 years per DROP score. DROP, Detecting Risk Of Postural hypotension; MMSE, Mini Mental State Examination.

postural hypotension.[50 51] Arterial stiffness is a postulated cause of interarm difference[52] and is also associated with postural hypotension[53 54]; thus interarm difference as a proxy measure of arterial stiffness might account for the observed association. Hypotension on ambulatory monitoring and elevated pulse wave velocity are both associated with cognitive decline, lending further support to the association of interarm difference, arterial stiffness and postural hypotension.[55]

Although postural hypotension is associated with diabetes, and with other complications such as neuropathy, retinopathy and proteinuria,[56] there was no univariable association in this study. Prevalence of postural hypotension in diabetes is associated with complications and duration of disease[57 58]; in this cohort, diabetes was present in only 6% of participants, whereas recent data suggest that 25% of adults over the age of 65 in the USA have it.[59] Therefore, a validation of our models in other larger representative populations is needed.

Postural hypotension has been associated with mild cognitive impairment.[60 61] and reduced cognitive performance.[62] Postural hypotension did not predict cognitive decline in a 2-year prospective study of older Finns[63] but is predictive over longer follow-up.[64] In the current analysis, postural hypotension per se was not predictive of cognitive decline over 9 years of follow-up but the DROP score was. This seems plausible given that it includes a number of risk markers known to be associated with cognitive decline.

### Relevance to clinical practice
Testing sitting (or lying) and standing blood pressure takes time and training. The skills of nurses measuring postural hypotension are variable when compared with guidelines[65]; incorrect arm positioning can underestimate postural

hypotension,[66] and the alerting reaction can overestimate it.[67] Early and accurate detection of postural hypotension is a prerequisite to intervening with medication withdrawal to reduce postural blood pressure drops and their associated risks including falls. Currently symptoms appear to be the main trigger for testing.[29] This should continue, however, a tool to identify which asymptomatic patients to test may help to target additional testing to those most likely to benefit. A DROP score of 1 or more appears to have such potential and may support proposals that individuals at elevated risk of postural hypotension should be tested.[68]

The strongly cardiovascular composition of the DROP score means that patients will commonly be taking antihypertensive drugs. Potential adverse effects of withdrawing antihypertensive medication to ameliorate postural hypotension are unclear, and medication withdrawal may concern clinicians, carers and patients. Risk of falls rises incrementally with each added orthostatic drug.[69] Prevalence of postural hypotension in hypertension is related to use of cardiovascular drugs (antihypertensive agents, vasodilators, diuretics),[70 71] alpha blockers[72] and the number of antihypertensive drugs used[51 73] and is associated with resistant or uncontrolled hypertension.[74 75] Successful treatment of blood pressure in the elderly is in fact associated with lower prevalence of postural hypotension,[76 77] but withdrawal of antihypertensive therapy improves postural hypotension.[78 79]

We retained Parkinson's disease in our models due to the strength of the association with postural hypotension, however, on clinical grounds, testing for postural hypotension would be better regarded as integral to any review in Parkinson's disease, given the high prevalence of postural hypotension in this condition.[49]

We sought to develop a pragmatic score to support busy clinicians, faced with a rising workload and increasingly multimorbid caseload.[31] Although measurement of blood pressure in both arms has become more frequent over time, it is not part of a routine review.[29 80] Therefore, we derived a DROP score omitting interarm difference, which performed with similar sensitivity and specificity. For the same reasons, we prefer the unweighted score as a practical aide memoire to recognition of the risk of postural hypotension.

### Further research

This study has examined the feasibility of identifying who should be tested with sitting and standing blood pressure measurements to detect asymptomatic postural hypotension. It seems that a simple pragmatic scoring system can support this. We need to refine and externally validate this approach in larger samples more representative of UK primary care. Further work is needed to examine the feasibility and implications of medication review and antihypertensive withdrawal based on detection of postural hypotension in primary care.

### CONCLUSION

We have described the derivation and validation of a score predicting the presence of postural hypotension. Initial testing suggests this approach to be feasible and has identified the potential use of the score in predicting mortality and cognitive decline over a 9-year period of follow-up. Further validation of the score in larger cohorts of individuals is warranted.

#### Author affiliations
[1]Primary Care Research Group, Institute of Health Research, University of Exeter Medical School, Exeter, UK
[2]Mental Health Research Group, Institute of Health Research, University of Exeter Medical School, Exeter, UK
[3]Longitudinal Studies Section, National Institute on Aging, Baltimore, Maryland, USA

**Contributors** CEC conceived and undertook this analysis. DT contributed to the analysis. FCW contributed to the analysis and offered statistical advice and support. DJL offered advice on analysis and interpretation of cognitive impairment indices. LF supported the study on behalf of the InCHIANTI investigators. JLC supervised study conduct. CEC drafted the manuscript, all authors revised and edited the manuscript and all authors have read, reviewed and approved the final manuscript.

**Funding** CEC is supported by an NIHR Clinical Lectureship award. DJL is funded by the Alzheimer's Society, the James Tudor Foundation, the Mary Kinross Trust, the Halpin Trust, and the National Institute for Health Research (NIHR) Collaboration for Leadership in Applied Health Research and Care South West Peninsula. This work was supported in part by the Intramural Research Program of the National Institute on Aging, National Institute of Health, Baltimore, MD 21224, USA.

**Disclaimer** The views expressed are those of the authors and not necessarily those of the NIHR, the NHS or the Department of Health.

**Competing interests** None declared.

**Patient consent** Not required.

**Ethics approval** The Italian National Research Council on Aging Ethical Committee approved the InCHIANTI study.

**Provenance and peer review** Not commissioned; externally peer reviewed.

**Data sharing statement** The InCHIANTI datasets are available on application with a research proposal to the InCHIANTI investigators at http://inchiantistudy.net/wp/.

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
