## [Reviewer comments · BMJ Open]

ARTICLE DETAILS

TITLE (PROVISIONAL)	Detecting Risks Of Postural Hypotension (DROP): derivation and validation of a prediction score for primary care
AUTHORS	Clark, Christopher; Thomas, Daniel; Warren, Fiona; Llewellyn, David; Ferrucci, Luigi; Campbell, John

VERSION 1 – REVIEW

REVIEWER	ANGELO SCUTERI DEPTM OF MEDICINE, SURGERY, AND EXPERIMENTAL SCIENCES, UNIVERSITY OF SASSARI, SASSARI (ITALY)
REVIEW RETURNED	11-Dec-2017

GENERAL COMMENTS	Authors test their research hypothesis in a large population with an adequate follow-up and with an original study design. Few points deserve clarification- - literature on postural hypotension and systemic hypertension / antihypertensive treatment is not updated. Several publications highlighted the frequent occurrence of hypotension in older subjects and its role in cognitive decline (for instance Int J Cardiol 2013).. Given that postural hypotension may reflect low BP levels, why SBP and DBP or MBP have not been included as covariates in the statistical models? (the same applies to age, in light of the significant older age reported in subjects with postural hypotension). One possible link between hypertension and postural hypotension is medical overtreatment in older subjects. However, the study does not report use of diuretics and of antihypertensive medication classes in the subgroups of volunteers. SCORE: there are discrepancies between definition and reported data. For instance, None showed a score equal to 4 in Figure 3, whereas it is reported in Figure 2 (mortality). How has the score been created? the variables from the predictive models (from 1 to 3) reported in Table 4 differ by number and by name. Why a specific predictive model has been preferred as compared to another?
---

REVIEWER	Stephen Juraschek
-----------------	-------------------

	Beth Israel Deaconess Medical Center, Boston, MA, USA
REVIEW RETURNED	13-Dec-2017

GENERAL COMMENTS	Review of “Detecting Risks Of Postural Hypotension (DROP): derivation and validation of a prediction score for primary care” Journal: BMJ Open Manuscript ID bmjopen-2017-020740 This is a well-written and well-performed study that develops a reasonable score with validation for predicting postural hypotension in a population of older adults. The biggest limitation pertains to the research question. Postural hypotension is arguably less important to prevent than its long-term sequelae (falls, syncope, stroke, dementia, etc). A score would be more clinically meaningful if it focused on these end points rather than postural hypotension. Also, it should be noted that the score requires an assessment of hypertension, while the full score also requires a BP assessment in both arms. In many cases, hypertension status is not known at the time of a visit. Postural hypotension requires at least two BP assessments as well with emerging evidence that early assessments are useful. Thus, the effort to determine the DROP score versus perform a direct assessment of postural hypotension seems comparable. In this sense, the DROP score would be more useful among patients where no BP measurements were necessary such that it could be derived from an electronic record to alert a provider to perform an OH assessment. In the Discussion the authors mention that orthostatic symptoms should continue to prompt screening for postural hypotension. Did the authors have data on orthostatic symptoms and whether the DROP score identified people with PH that were asymptomatic? Were people with symptoms excluded from the study? Minor: Please repeat the final model (results of backwise regression) in Results text to avoid need to reference table
---

VERSION 1 – AUTHOR RESPONSE

bmjopen-2017-020740 – response to editorial and reviewer comments

13-Dec-2017

Editorial Requests:

- Can the 'Strengths and Limitations' section on page 3 be shortened? The section should ideally contain up to five short bullet points, no longer than one sentence each, that relate specifically to the

methods or design of the study reported (see: <http://bmjopen.bmj.com/site/about/guidelines.xhtml#articletypes>). It should not be a summary of the study and its findings, so the final bullet point needs revising or removing.

Bullet points have been revised and shortened as suggested

- Please add a statement to the methods section confirming that written informed consent was obtained from participants for the InCHIANTI study.

This statement has been added to paragraph 2 – methods as requested

Reviewers' Comments to Author:

Reviewer: 1

Reviewer Name: ANGELO SCUTERI

Institution and Country: DEPTM OF MEDICINE, SURGERY, AND EXPERIMENTAL SCIENCES, UNIVERSITY OF SASSARI, SASSARI (ITALY) Competing Interests: NONE

Authors test their research hypothesis in a large population with an adequate follow-up and with an original study design.

Few points deserve clarification-

- literature on postural hypotension and systemic hypertension / antihypertensive treatment is not updated. Several publications highlighted the frequent occurrence of hypotension in older subjects and its role in cognitive decline (for instance Int J Cardiol 2013)..

Thank you for this point. We do cite a number of references, the most recent being 2009, concerning the prevalence of postural hypotension with hypertension. Thank you for recommending your paper, which deals with hypotension detected on ambulatory monitoring. We are pleased to cite it in our discussion.

Given that postural hypotension may reflect low BP levels, why SBP and DBP or MBP have not been included as covariates in the statistical models? (the same applies to age, in light of the significant older age reported in subjects with postural hypotension).

Thank you for commenting on this; we did include age in the analysis. Use of blood pressure variables in the models made no difference to the outputs apart from replacing “presence of hypertension” with systolic blood pressure. We have amended table 3 to include baseline blood pressure comparisons, and describe the sensitivity analysis of blood pressure terms vs presence of hypertension in the results.

One possible link between hypertension and postural hypotension is medical overtreatment in older subjects. However, the study does not report use of diuretics and of antihypertensive medication classes in the subgroups of volunteers.

Thank you for highlighting this. Individual blood pressure drug classes were included in the analyses; they did not contribute to the final models and were unintentionally omitted in writing up. These data now appear in Table 3, with relevant changes in the results. Table 3 has been revised to include previously omitted percentages.

SCORE: there are discrepancies between definition and reported data.

For instance, None showed a score equal to 4 in Figure 3, whereas it is reported in Figure 2 (mortality).

As described in the results only one individual in the validation cohort attained a score of 4. That participant did not have mini-mental stated examination data. We have addressed this limitation in discussion and in the revised strengths and limitations section. Results section now clarifies the proportion with MMSE data available.

How has the score been created? the variables from the predictive models (from 1 to 3) reported in Table 4 differ by number and by name.

The only difference between the three models is in the inclusion or exclusion of the term for inter-arm difference. This is included as a continuous variable in model 1, categorical in model 2 and not included in model 3.

Why a specific predictive model has been preferred as compared to another?

We describe in results where simpler and more pragmatic scores were preferred and selected for simplicity.

Reviewer: 2

Reviewer Name: Stephen Juraschek

Institution and Country: Beth Israel Deaconess Medical Center, Boston, MA, USA Competing

Interests: None

This is a well-written and well-performed study that develops a reasonable score with validation for predicting postural hypotension in a population of older adults. The biggest limitation pertains to the research question. Postural hypotension is arguably less important to prevent than its long-term sequelae (falls, syncope, stroke, dementia, etc). A score would be more clinically meaningful if it focused on these end points rather than postural hypotension.

We agree with this statement and recognise the evidence to associate postural hypotension with these important outcomes, indeed we do report the score's associations with survival and cognitive prospective outcomes. However our experience and survey findings suggest that postural hypotension is not sought in the absence of symptoms. We contest that it cannot be managed without being detected, hence our study hypothesis that a score, if feasible, might promote detection of postural hypotension in clinical assessment, thus offering the opportunity to review medication and offer advice to ameliorate postural hypotension and future sequelae.

Also, it should be noted that the score requires an assessment of hypertension, while the full score also requires a BP assessment in both arms. In many cases, hypertension status is not known at the time of a visit. Postural hypotension requires at least two BP assessments as well with emerging evidence that early assessments are useful. Thus, the effort to determine the DROP score versus perform a direct assessment of postural hypotension seems comparable. In this sense, the DROP score would be more useful among patients where no BP measurements were necessary such that it could be derived from an electronic record to alert a provider to perform an OH assessment. In the Discussion the authors mention that orthostatic symptoms should continue to prompt screening for postural hypotension.

We understand your statement, however we have shown (and reference our findings) that direct assessment of postural hypotension does not in practice take place unless symptoms are reported. That is why we have set out to identify those most likely to benefit from testing for postural hypotension.

Did the authors have data on orthostatic symptoms and whether the DROP score identified people with PH that were asymptomatic? Were people with symptoms excluded from the study?

Our outcome measure was postural hypotension per se not postural symptoms, since we sought to predict the postural drop itself. We do state that we agree with current guidelines recommending testing in the presence of symptoms, and have confirmed that this happens already in practice. The relevance of the score is, therefore, in recognising who to test in the absence of symptoms.

Minor: Please repeat the final model (results of backwise regression) in Results text to avoid need to reference table

Thank you – this has been added

VERSION 2 – REVIEW

REVIEWER	ANGELO SCUTERI DEPTM MEDICINE, SURGERY AND EXPERIMENTAL MEDICINE UNIVERSITY OF SASSARI ITALY
REVIEW RETURNED	08-Jan-2018

GENERAL COMMENTS	Comments have been adequately addressed.
--

REVIEWER	Stephen Juraschek Beth Israel Deaconess Medical Center/Harvard Medical School
REVIEW RETURNED	21-Jan-2018

GENERAL COMMENTS	I think it would be helpful to see how many people symptomatic and asymptomatic people were identified by the score. Can the authors provide these data?
--

VERSION 2 – AUTHOR RESPONSE

Dear Dr Sucksmith

Thank you for accepting the revised manuscript. We understand the reviewer's suggestion regarding symptoms of postural hypotension however we do not have specific data on postural symptoms, only falls in the year preceding each follow up. We have therefore made three further minor revisions to the manuscript as follows:

1. A revision to the symptoms statement in the Strengths and Limitations bullet points.
2. The addition of a line on the associations of the DROP score with previous falls in results section.

3. We have acknowledge our lack of postural symptom data in the strength and weaknesses section of the discussion.

I trust that these revisions meet your requirements.

Yours sincerely
Chris Clark

VERSION 3 – REVIEW

REVIEWER	Stephen Juraschek Beth Israel Deaconess Medical Center/Harvard Medical School, USA
REVIEW RETURNED	16-Feb-2018
GENERAL COMMENTS	The authors have adequately responded to my comments.